# Distribution of DC Subtypes: CD83+, DC-LAMP+, CD1a+, CD1c+, CD123+, and DC-SIGN+ in the Tumor Microenvironment of Endometrial Cancers—Correlation with Clinicopathologic Features

**DOI:** 10.3390/ijms24031933

**Published:** 2023-01-18

**Authors:** Grzegorz Dyduch, Apolonia Miążek, Łukasz Laskowicz, Joanna Szpor

**Affiliations:** 1Department of Pathomorphology, Faculty of Medicine, Jagiellonian University Medical College, Grzegorzecka 16, 31-351 Krakow, Poland; 2Gynaecology and Oncology Clinical Department, University Hospital, Jakubowskiego 2, 30-688 Krakow, Poland

**Keywords:** endometrial cancer, dendritic cells, immunohistochemistry, cancer immunotherapy

## Abstract

Treatment options for endometrial cancer (EC) do not provide satisfactory survival improvement for advanced cases, hence the interest in novel therapies utilizing immunological regulatory mechanisms. Measures to modify the functionality of dendritic cells (DCs) found in TME are intensively investigated, given that DCs play a crucial role in inducing antitumor immunity. Samples of malignant endometrial neoplasms obtained from 94 patients were immunohistochemically stained with selected antibodies. Counts of positively identified DCs were correlated with clinical advancement and histological malignancy of cancers. The most prominent DC subtypes were immature DC-SIGN+ or CD123+. Mature CD83+ DCs were the fewest. We found a significant divergence of grade value distribution between cancers of different DCs’ CD1a+ counts. The DC-LAMP+ count was positively associated with grade. Cancers with the least DC CD1c+ or DC CD123+ had higher pT scores than ones that were more heavily infiltrated. ECs can suppress immune cells, hence the predominance of immature DCs in our samples. Associations between DC counts and clinicopathological features of EC were observed only for a few subsets, which was plausibly due to the low diversity of the obtained samples or the small group size. Predictive abilities of particular DC immune subsets within EC’s TME remain ambiguous, which calls for further research.

## 1. Introduction

The term “tumor microenvironment” (TME) refers not only to cells and extracellular matrix, but also to signaling factors delivered from cancerous and infiltrating cells. These factors play a crucial role in intercellular interactions, and their composition varies significantly as the neoplasm grows and invades the surrounding tissues [1,2]. The TME’s immunological component comprises cells belonging to both innate and adaptive immune systems with dendritic cells (DCs) acting as a connector between them [3]. DCs’ capacity for efficient antigen cross-presentation to CD8+ lymphocytes that induces antineoplastic activity has contributed to an increased interest in their part in response to currently used treatments [4]. Moreover, therapies directly using DCs (ex vivo generated DCs) or ones that improve their functionality in vivo are being evaluated for fighting different cancers [5]. Recent studies that focused on complex interactions between tumor cells and their environment described many tumor-derived immunosuppressive factors that cause dendritic cell dysfunction. These studies shed light on pathways whose modification might potentially restore DCs’ functionality in cancer surroundings [5,6]. Endometrial cancer (EC) is the third most common cancer in females worldwide and the fourth-leading cancer-related cause of death amongst Polish women [7]. Different immune subtypes of EC have been recognized and correlated with disparities in patients’ survival. For instance, high densities of macrophages, DCs, and lymphocytes have been associated with a more favorable prognosis [3]. Despite a rising interest in DCs in oncological therapies, their contribution to anti- or pro-tumor immune activity in endometrial cancers is not fully understood [8]. Some papers proved that high DC scores are related to better survival, while others claimed that the increasing DC infiltration is positively associated with cancer stage, grade, and poor outcomes [9,10,11]. Inconclusive results often stem from differences in surface markers and intracellular molecules selected by researchers to distinguish subtypes of DCs. In the present study, we aimed to assess different subsets of dendritic cells in endometrial cancer samples using IHC staining pattern analysis. We hypothesized that the broader array of antibodies used (anti-CD83, -DC-LAMP, -CD1a, -CD1c, -CD123, and -DC-SIGN) would clarify the correlation between DCs’ infiltrates and the clinicopathological features of endometrial cancers. 

## 2. Results

### 2.1. Patient Characteristics

Patients from whom samples were obtained constituted 94 women aged from 40 years old (y.o.) to 86 y.o. (median age: 64 y.o.) at the time of diagnosis (Table 1 and Appendix A). The majority were diagnosed with endometroid adenocarcinoma. Other malignant neoplasms of the uterus (carcinosarcoma, clear cell, mucinous, and mixed carcinoma) were identified only in individual cases. Due to unexpectedly small representations of non-endometrioid cancer types in the studied population, analyses of relationships between their clinicopathological features and DC counts could not be performed (Appendix A). As a result, statistical tests were established for endometrioid adenocarcinomas only. The studied cancer samples were predominantly early-staged: most cases were marked as grade 1 or grade 2, and low-stage cancers prevailed (Table 1).

### 2.2. Counts of Different DC Subsets

Overall, we observed that the tumor stroma was more heavier infiltrated by DCs than its invasive margin (Figure 1, Figure 2, Figure 3 and Figure 4; Table 2). However, differences were statistically significant only for DC-SIGN+ (*p* = 0.0278), DC CD1a+ (*p* < 0.00001) and DC-LAMP+ (*p* = 0.0061). For all examined DC populations, we noticed significant strong correlations between their counts in the tumor stroma and margin (Figure 5). Moreover, some other interesting correlations were found, such as between the number of DC-CD83+ and DC-CD1a+ on the cancer’s border (r = 0.464078; *p* < 0.05) or between DC-CD1a+ and DC-CD1c+ in the tumor’s stroma and glands (r = 0.395171 and 0.342721, respectively; *p* < 0.05). Within endometroid cancers’ stroma, the most prominent were DC-SIGN+ (median number: 74; interquartile range (IQ): 40–134) followed by CD123+ (Me: 53; IQ: 26–89) (Figure 4B). Likewise, the two mentioned subtypes were the most numerous in cancers’ invasive margins (Me: 57, IQ: 22–95 for DC-SIGN+; and Me: 49, IQ: 26–103 for CD123+) (Figure 4A). Only DC CD1a+ and CD1c+ infiltrated adenocarcinomas’ glandular epithelium, the former in higher numbers (Me: 75; IQ: 41–119) (Figure 4C). For all evaluated areas, the DC populations were statistically greater in adenocarcinomas than in healthy endometrium (Figure 6A–C; Table 3).

### 2.3. Statistical Analyses: Correlations between DC Counts and ECs’ Clinicopathological Features

In further analyses, we compared the groups created for each IHC staining based on the median, Q1, and Q3 values of the DC count. This method of allocation allowed for even numbers of samples in each group, whereas raw DC count (/mm^2^) comparisons between cancers grouped based on their pathological grade values or pT or pN scores could have been biased due to the unequal group sizes. Indeed, the statistical analyses performed for these comparisons showed no significant results (Appendix A). We found statistically significant differences in grade value distribution between four groups of ECs that were divided based on the amount of DC CD1a+ present in the invasive margin (Figure 7A). Groups of cancers infiltrated by DCs positively identified for other antibodies were not significantly different concerning histological differentiation (Table 4). We observed some diversity in patients’ ages between four groups of ECs distinguished by the numerosity of DC CD1a+ in glandular epithelium. The youngest patients were represented predominantly in the group of cancers with the least numerous DCs (Figure 7B). Another observation was that cancers with the least DC CD1c+ present in the stroma (DC group 2) infiltrated surrounding tissues to a greater degree than ones most heavily infiltrated by CD1c+ cells (DCs group 4) (Figure 7C). Similarly, cancers with the highest observed pT were predominant in the DC CD123+ group 2 (Figure 7D). There were no significant differences between cancers of the different stages (pT and pN) regarding counts of other DC subsets in intrastromal, marginal, or glandular locations (Table 4). 

## 3. Discussion

Dendritic cells form a heterogenous cell population represented by many subsets. They are distinguished based on different transcription factors, surface molecules, and functions within the cancerous microenvironment. The most stable and explicit expression of CD83, a surface molecule that belongs to the Ig superfamily, characterizes mature DCs. However, CD83 specificity is not absolute because it can be found on the surface of other activated immune cells such as T cells, macrophages, and B cells [12,13]. DC-LAMP (CD208), another marker of mature DCs and a lysosome-associated membrane glycoprotein, exhibits greater specificity for DCs [14]. CD1a and CD1c represent transmembrane glycoproteins that are homologous with MHC molecules but are involved in the presentation of non-peptide antigens (lipids and glycolipids) [15,16]. Both are regarded as immature DC markers. However, CD1c+ DCs represent the most numerous subset of conventional DCs (cDCs) found in human peripheral blood, which plays a crucial role in inducing antitumor immunity [5,17]. DC-SIGN (CD209) is a transmembrane C-type lectin that facilitates binding to glycoconjugates and the internalization of pathogens. It is expressed in particular on monocyte-delivered DCs (MoDCs) and skin DCs, thereby enabling adhesion to, amongst others, cancer cells [18]. CD123 is an IL-3 receptor alpha chain expressed by immature plasmacytoid DCs [19,20].

Endometrial cancer is considered immunogenic due to its selected molecular subtypes, especially POLE-ultramutated and MSI-hypermutated, which are associated with a high infiltration of inflammatory cells [21,22]. The positive impact of tumor-infiltrating immune cells on EC prognosis is well established, but the predictive abilities of particular immune subsets are more ambiguous [22]. Unified and holistic knowledge of the composition and functions of these subsets would facilitate the design of immunotherapy agents for EC treatment [21,22,23,24]. The role of dendritic cells in antitumor immunity has been exhaustively explored and is unargued in the context of recent findings [25,26,27,28,29]. Nowadays, DCs are considered potent APCs that are crucial for activating T cells and conditioning the tumor microenvironment (TME) with different cytokines. Presumably, the most comprehensively studied subset in the context of antitumor immunity is conventional DCs (cDCs)—cDC1s in particular. After capturing cancer antigens, DCs mature and migrate to the tumor-draining lymph nodes (LNs), where they present antigens to T cells, thereby initiating the recruitment of T cells into the TME. Dendritic cells can also directly interact with naive and effector T cells within the TME [28]. Factors released upon the destruction of cancerous cells by chemotherapeutics enhance DC activation and antitumor CD8+ T-cell responses [28,29]. However, the TME considerably impacts dendritic cell functionality. It contains many immunosuppressive factors such as VEGF, IL-6, IL-10, PGE2, and LXRalfa that are delivered from cancerous and other residual cells [1,15] that limit DCs’ migratory capacity and inhibit their maturation as well as their efficient antigen presentation [17,30]. Neoplasms trap dendritic cells by hindering their capabilities to travel to LNs and promoting apoptosis through FasL and TRAIL [30]. Other established mechanisms by which cancers influence DCs include natural killer cell (NK) inhibition (e.g., by tumor-delivered PGE2) [5,22]. NK cells produce CCL5 and XCL1, which act as chemoattractants for DCs, thereby contributing to their intratumoral accumulation. NK cells are also the primary source within the TME of FLT3L [5,28], a crucial factor for DCs’ maturation and functionality. Those and other mechanisms that are not fully understood may be the reasons for the low abundance of mature DCs within the TME. 

The results of our study showed that the most numerous DCs present in both the tumor stroma and margin were immature ones that expressed the DC-SIGN molecule, which was in agreement with the former statements. Immaturity of DCs resulted in the ineffective presentation of tumor antigens that induced tolerance toward them and promoted the persistence of malignancy [31,32]. Indeed, there have been reports of an association between a high number of immature DC-SIGN+ cells and angiolymphatic invasion by cancerous cells [33]. Further, immature DC-SIGN+ DCs within colorectal cancer (CRC) stroma facilitated CRC escape from immune surveillance and poor prognosis [34,35]. However, we found no significant associations between the counts of DC-SIGN+ cells and the advancement of endometrial cancer. Notably, there were other sources of DC-SIGN expression, including monocyte-derived DCs and activated macrophages [18,31], which might have led to a possible overstatement of the mean counts of DCs positively identified for DC-SIGN. This could explain the relatively high counts of DC-SIGN-positive cells in the healthy endometrium used as an internal control. 

The least represented cells in the examined samples were those positively identified for CD83, a marker for fully mature DCs [36]. Upon proinflammatory activation (e.g., through toll-like receptor engagement), CD83 is transported to the cell membrane, thereby promoting MHC-II stabilization [12]. Indeed, APCs with high CD83 expression exhibit MHC II and CD86 upregulation thanks to the ability of the CD83 molecule to negatively impact the activity of the ubiquitin ligase MARCH-1 [12,37]. Unexpectedly, recent studies on CD83-deficient DCs showed that the lack of this molecule led to the enhancement of immune responses. DCs insulated from CD83 KO mice expressed higher amounts of IL-2, CD23, and OX40L, which resulted in more potent induction of T-cell responses (despite reduced MHC-II expression) and suppression of Treg lymphocytes [38]. Studies on different neoplasms have shown that the density of CD83+ cells was lower in cancerous tissues compared to healthy controls, and it decreased as the disease advanced [39]. We obtained the opposite results and showed that the mean numbers of CD83-positive cells and of all examined DCs subtypes were higher in endometroid cancers than in healthy controls. Many authors observed negative associations between DC83+ cell counts and tumor size or metastatic occurrence [9,15] and claimed that the density of DCs CD83+ within cancerous tissue was well suited for predicting survival [39,40]. However, in our population, no significant correlations between CD83+ cells and clinicopathological features of EC were found. This might have been due to the low diversity of obtained samples (the majority were marked as grade 2, stage pT1) or the small group size. 

Cells positive for another maturation marker—DC-LAMP—were also one of the least represented in our samples. Studies regarding melanomas reported that a high density of CD208+ cells in sentinel lymph nodes was associated with prolonged survival of patients [14]. Likewise, Ludovic Martinet et al. showed positive correlations between counts of DC-LAMP+ dendritic cells and T-cell infiltration within breast cancer tissues. The presence of DC-LAMP+ cells was also associated with favorable outcomes for these patients [41]. We did not find significant correlations between the count of DC-LAMP+ cells and cancer grade or stage scores. However, we noticed a tendency for higher-grade cancers to have greater densities of DC-LAMP+ DCs infiltrates. Less-differentiated tumors usually have a higher mutation burden [21] and an increased immunogenic potency that lead to higher DC-LAMP+ cell accumulation [42,43,44]. However, neither cancer grade and count of mutations nor mutational burden and infiltration with immune cells have been linked directly. More research on larger groups is needed to support the above statements because this study failed to prove significant differences in the DC-LAMP+ cell count between cancers with different grade scores or stage scores.

CD1a+ and CD1c+ DCs were the only subsets that infiltrated ECs’ glandular epithelium. In agreement with other authors, we showed that expression of CD1a was stronger in endometrial cancer’s stroma and glands than in healthy endometrium [10]. Several reports on many neoplasms, including thyroid [45] and gastric cancer [46], proved that the infiltration of immature CD1a+ cells was associated with favorable clinical outcomes. In oral squamous cancer samples, lymph node metastases co-existed with a significant depletion in stromal CD1a+ dendritic cells [47]. A study on mycosis fungoides (MF) proved that the reduced presence of CD1a+ populations (DCs and Langerhans cells) was associated with a resistance to therapy [48]. These unexpected results (because DCs that express CD1a and CD1c are considered immature) may come from CD1a involvement in non-peptide antigen presentation by DCs [45]. Given that lipid compositions change significantly throughout carcinogenesis [15], efficient presentation of lipid and glycolipid antigens to T cells could potentially promote antitumor activity. In this study, cancers with the least numerous DC CD1a+ on their infiltrative borders had statistically higher grade scores than ones with higher DC counts. This observation could have resulted from the fact that ECs with higher grade scores were less likely to form glandular structures, to which DCs CD1a+ showed a great affinity. It is also possible that a higher number of DCs CD1a+ in cancerous glands induced effective immune responses that controlled the EC’s progression and dedifferentiation, hence the lower grade scores. Counts of DC CD1a+ were not correlated with EC’s pathological stage regardless of the evaluated distribution, which was consistent with the results of a similar IHC research paper on EC [10]. We observed that the youngest patients were represented predominantly in the EC group of the least numerous DC CD1a+ present in tumors’ glandular epithelium. Patients’ ages in this group were significantly lower than in cancers with higher counts of CD1a-positive cells in EC glands (group 3). Generally, an older age is associated with functional and structural changes in innate immune cells and higher concentrations of proinflammatory cytokines that facilitate the activation and maturation of DCs [49]. However, it is also directly related to a significantly lower density of DC CD1a+ in mucosal or skin epithelium. A lower expression of CD1a+ was expected in older patients. However, in the EC groups 1 to 3, there was a tendency for DC CD1a+ numerosity to increase along with age. Contrary to this, cancers with the most numerous DC CD1a+ (group 4) in the glandular epithelium comprised relatively young patients. An inconsistency between these two observations may have resulted from the small group sizes or the large diversity of patients’ ages in the four groups. Another explanation is the possible difference in percentages of the glandular epithelium—linked with DC CD1a+ numerosity—between cancers from different groups. There were no statistically significant differences in the EC’s grade (or stage) between patients of different ages, so clinicopathological features could not fully explain the observed variation in the CD1a+ expression between the groups. 

A typical way of dividing DCs distinguishes conventional DCs (cDC1s or cDC2s), plasmacytoid DCs (pDCs), and monocyte-delivered DCs (MoDCs). The CD1c molecule is expressed on the surface of cDC2s—presumably, the most prominent DC subset found in human blood [17,49]. It is, however, a cCD1 subset amongst all others that best correlates with favorable patient survival when present within tumor tissues. cDC2s orchestrate immunity to extracellular pathogens by presenting antigens via MHC-II to CD4+ helper T cells and are less potent in inducing proinflammatory CD8+-mediated responses [28]. We found a significant difference in the EC infiltration depth between the DC CD1c+—high group (group 4) and a group of lower CD1c+ density (group 2). Patients with the highest intrastromal DC CD1c+ density had the most cancers with low pT scores. This observation suggested that cDC2s, despite being less potent APCs than cDC1s, still associate with favorable features of ECs such as a lower cancer stage, which arguably is due to migratory cDC2s’ capacity to drive effective CD4+ T-cell responses by priming naive cells in tumor-draining LNs [28].

We did not find an explanation in the literature for the presence of DCs expressing molecules from the CD1 family amid cancerous glandular epithelial cells. Plausibly, neoantigens presented by these cells are specific to cancerous glands and hence could constitute tumor-specific antigens used for loading onto DCs in cancer immunotherapy [21].

The CD123 molecule is expressed predominantly on immature plasmacytoid DCs (pDCs) [19,50], which seem to contribute to antitumor responses to a lesser degree than conventional DCs [20]. Nevertheless, despite a lower antigen cross-presentation capacity, secretion of IFNα by pDCs is essential for cDC1 maturation and stimulation of local CD8+ T cells [28]. However, tumor infiltration by pDCs has been correlated with a poor prognosis for patients with various cancers, supposedly due to their impaired response to TLR7/9 activation and decreased IFN-α release, thereby contributing to the production of IL-10, TGF-β, and Treg cells [51,52,53]. In contrast, we observed that the EC group with the highest infiltration of DC CD123+ in the invasive borders had significantly more cases with a lower cancerous infiltration depth compared to a group with decreased DC CD123+ density. Potentially, not all cancers cause pDC hypofunction or cause it to various degrees, which would explain observations that some pDCs responded to signaling through TLR7/TLR8. Further, Stefan Nierkens et al. proved that the response to cancer immunotherapy depended on cross-talk between pDCs and cDCs [54].

The most common EC histological subtype is endometroid carcinoma, which accounts for up to 90% of cases [22]. It has a better prognosis than any other type, including clear cell (CCC), serous, and mucinous cancers [28]. Over 80% of endometroid subtype cases present molecular aberrations in the PI3K–PTEN–AKT–mTOR pathway. Additionally, many represent MSI-positive or POLE-mutated cancers, which are effectively recognized by immune cells due to high neoantigen loads [55]. Serous and mixed-histology tumors account for the majority of the copy-number high-molecular subgroup, which is characterized by a low expression of immune-related biomarkers and unfavorable clinicopathological features [21,22]. Serous carcinomas are considered FIGO grade 3 due to significant cytological atypia [55]. This histological type represents up to 10% of EC cases, but we did not recognize any serous carcinoma in our samples. Both serous and clear cell carcinomas often display mutations in the TP53 gene. However, the latter cannot be assigned to any specific molecular profile in most cases [21,55]. We had only one patient with CCC in which the highest counts of DC-SIGN+ and marginal CD83+ were observed. We noticed the most numerous mature DCs (expressing CD83 or DC-LAMP) in the mucinous carcinoma samples. More research on larger groups is warranted to determine whether high densities of mature DCs infiltrating this subtype are repetitive findings. Carcinosarcomas demonstrate mesenchymal differentiation and mutations in TP53, whereas POLE and MMR defects are rare [55]. Mean counts of CD123+ infiltrating both stroma and margins of two carcinosarcoma cases were the highest amongst all examined samples.

We noticed some inconsistencies in the results of studies that evaluated associations between DC subsets counts and clinicopathological features of neoplasms. These may have been a consequence of a significant disparity in the activation of the immune cells by antigens delivered from different tumors, especially since only a few cancers are highly immunogenic. Moreover, the diverse composition of antibodies used for IHC stainings impeded insight into the distribution of DC subsets within tumors. To our knowledge, this study was one of the few that thoroughly investigate the counts of that many DCs subsets within endometroid adenocarcinomas. Another reason for the incoherent results was the significantly different methodology of the mentioned studies. Even within ones based on an IHC analysis, the values used as cut-offs between low and high DC infiltration were varied. We chose the Q1, median, and Q3 values as split points to improve the specificity of our results. Finally, we hypothesized that different approaches that focused on the distinction between cDC1s, cDC2s, or plasmacytoid DC-infiltrating cancers rather than on the expression of individual DC molecules would have a better predictive value. We assumed that quantifying different DC subsets would be even more relevant if paired with staining for functional markers expression; e.g., PD-1. We plan to broaden our investigations and evaluate associations between densities of different DC subpopulations and specific molecular types of EC following TCGA molecular classification. Additionally, marking for PD1 and PD-1L on both cancerous and immune cells within TME will be conducted. 

## 4. Materials and Methods

### 4.1. Tissue Specimens

The materials analyzed in this study were 94 paraffin-embedded tissue samples from hysterectomies performed for malignant endometrial neoplasms between 2012 and 2014 that were obtained from the archives of the Department of Pathomorphology, Jagiellonian University Medical College, Krakow. Analysis of the pathological reports allowed for data extraction regarding the cancers’ histological malignancy (grade) and clinical advancement (stage), including the depth of cancerous invasion and metastases to lymph nodes. Additionally, information on patients’ ages was obtained and allowed for creating five age groups: 1st (patients aged <40 to 50); 2nd (aged <50 to 60); 3rd (aged <60 to 70); and 4th (aged <70 to 80); and 5th (aged ≥80 years old).

### 4.2. Immunohistochemistry

The chosen tissue samples were stained manually with six primary monoclonal antibodies: anti-CD83, -DC-LAMP, -CD1a, -CD1c, -DC-SIGN, and -CD123 in compliance with the protocols routinely used in our department. All staining procedures were in line with the typical protocols used in the field. Paraffin blocks from uterus tissues were sectioned at a 4 μm thickness and incubated at 34 °C for 12 h. Sections were then deparaffinized and dehydrated. The activity of endogenous peroxidase was inhibited by incubating the tissues with 3% hydrogen peroxide for 10 min. Antigen retrieval was performed by immersing the slides in citrate buffer (pH 6.0; 0.01 M) or EDTA (pH 8.0; 0.01 M) and subjecting them to 97 °C in a water bath for 30 min. 

Polyclonal secondary antibodies conjugated to horseradish peroxidase (Ultra Vision LP Value Detection System HRP Polymer, Lab Vision, Thermo Scientific, Waltham, MA, USA) were applied to visualize the obtained antigen–antibody complexes using DAB (3,3-diaminobenzidine) as the chromogen. Cell nuclei were stained with hematoxylin to enhance the contrast in the tissue sections.

### 4.3. Evaluation of Immunostaining

Quantitative analysis of immunohistochemical reactions was performed using light microscopy, and the numbers of positively stained cells for each antibody were obtained. Slides were initially examined at a low magnification (10× lens) to select the areas that were most abundant in positively identified DCs (so-called hot spots). Within the chosen areas, the counts of positively stained DCs in five representative fields of view were summed up at a high magnification (40× lens) and expressed per 1 mm^2^. An assessment of cells’ morphology preceded establishing their numerosity: a cell was considered a positively stained DC if it had a visible nucleus, dendritic appearance (e.g., cytoplasmic processes), and intensely colored cytoplasm at 40× magnification. Analyses of each monoclonal antibody’s reactivity were performed separately for the tumor stroma, glandular endothelium, and invasive border (Appendix A). The latter term refers to the field of view comprising the cancerous margin and the adjacent healthy tissues in an even ratio. Samples were also evaluated for the presence of co-occurrent non-neoplastic endometrium. Numbers of positively identified DCs per mm^2^ in stroma, glands, and margins of healthy tissue were utilized to create an internal positive control group (Appendix A). Two authors performed the pathological analyses separately and solved any disagreements via discussion.

### 4.4. Statistical Analysis

The software used for the data analysis were IBM SPSS Statistics 28 and Statistica 10 (StatSoft, Tulsa, OK, USA). For every antibody, samples of endometroid cancers were assigned to four groups that were distinguished based on the first quartile (Q1), median, and third quartile (Q3) values of the positively identified DC count as cut-offs. (Table 5) Differences between groups regarding dependent qualitative variables (cancer’s grade and stage) were evaluated using Kruskal–Wallis ANOVA (followed by a post hoc Dunn test) and Fisher tests. Additionally, a Kruskal–Wallis ANOVA test was used to perform comparisons of the raw (not divided into groups) DC counts (/mm^2^) between three grade scores (grades 1, 2, and 3), six pT scores (pT1, pT1a, pT1b, pT2, pT3a, and pT3b), and three pN scores (pN0, pN1, and pN2). As seen in Table 1, the numbers of samples allocated to grade or stage scores were unevenly distributed in the studied population just as they were in the different antibody subgroups, thereby increasing the risk of biased results. The relationship between the number of DCs and a patient’s age was evaluated based on the results of the Mann–Whitney U test or Kruskal–Wallis ANOVA test. Finally, the Spearman rank correlation test was utilized to search for correlations between the numbers of DC subsets in different locations within the tumor tissues; *p*-values < 0.05 were considered significant.

## Figures and Tables

**Figure 1 ijms-24-01933-f001:**
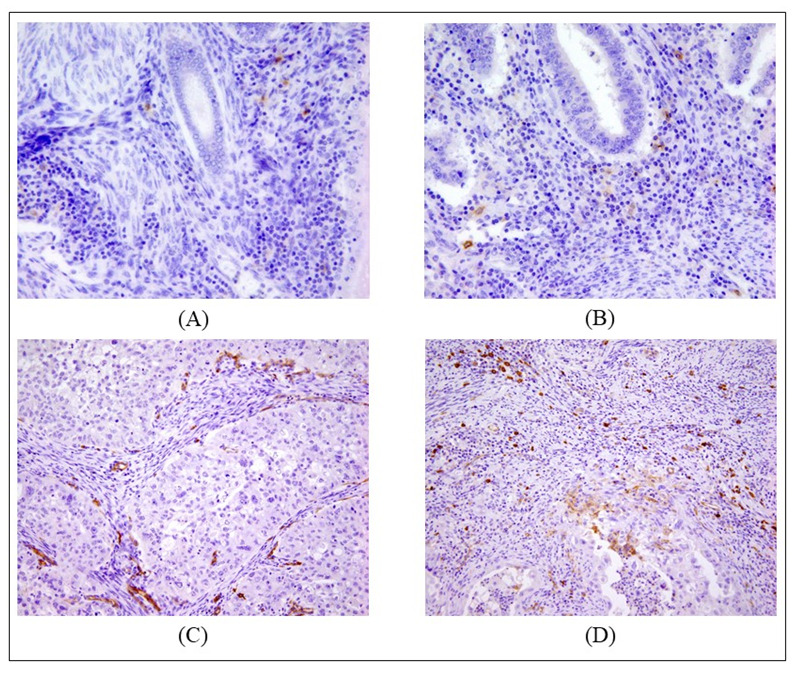
Examples of positively identified DCs infiltrating EC samples. A cell was considered as positively stained if at ×40 magnification it had a visible nucleus, dendritic appearance, and intensely colored cytoplasm. (**A**) DC CD83+ infiltrating stroma of endometroid cancer at 40× magnification (8 cells/high-power field (HPF)); (**B**) DC CD83+ infiltrating border of endometroid cancer at 40× magnification (12 cells/HPF) (**C**) DC CD123+ infiltrating stroma of endometroid cancer at 40× magnification (52 cells/HPF); (**D**) DC CD123+ infiltrating invasive border of endometroid cancer at 20× magnification. In (**C**,**D**) a sign of staining can be seen in the endothelium of intratumoral vessels.

**Figure 2 ijms-24-01933-f002:**
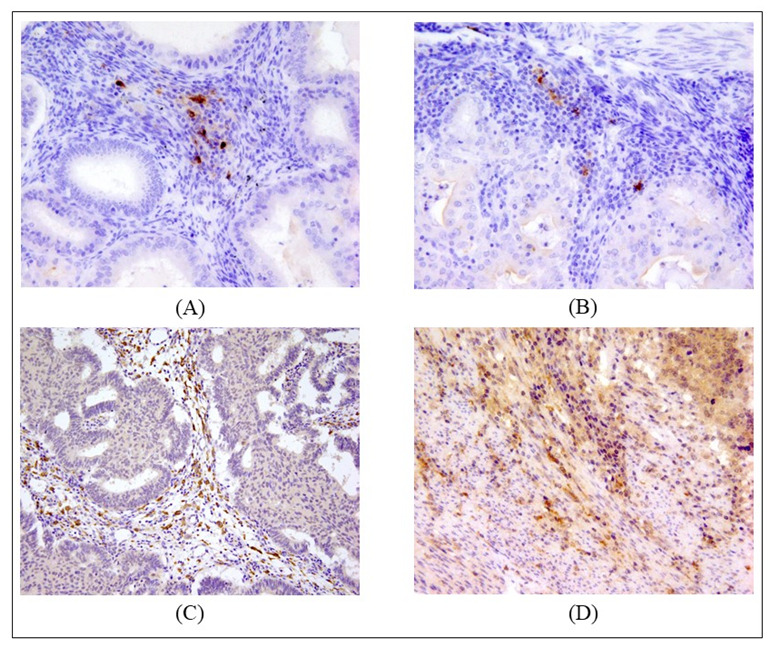
Examples of positively identified DCs infiltrating EC samples. A cell was considered as positively stained if at 40× magnification it had a visible nucleus, dendritic appearance, and intensely colored cytoplasm. (**A**) DC-LAMP+ infiltrating stroma of endometroid cancer at 40× magnification (7 cells/HPF); (**B**) DC-LAMP+ infiltrating border of endometroid cancer at 40× magnification (8 cells/HPF); (**C**) DC-SIGN+ infiltrating stroma of endometroid cancer at 20× magnification; (**D**) DC-SIGN+ infiltrating invasive border of endometroid cancer at 20× magnification. In (**C**,**D**) a sign of staining can be seen in the cancerous glands and off-target staining in stromal tissues.

**Figure 3 ijms-24-01933-f003:**
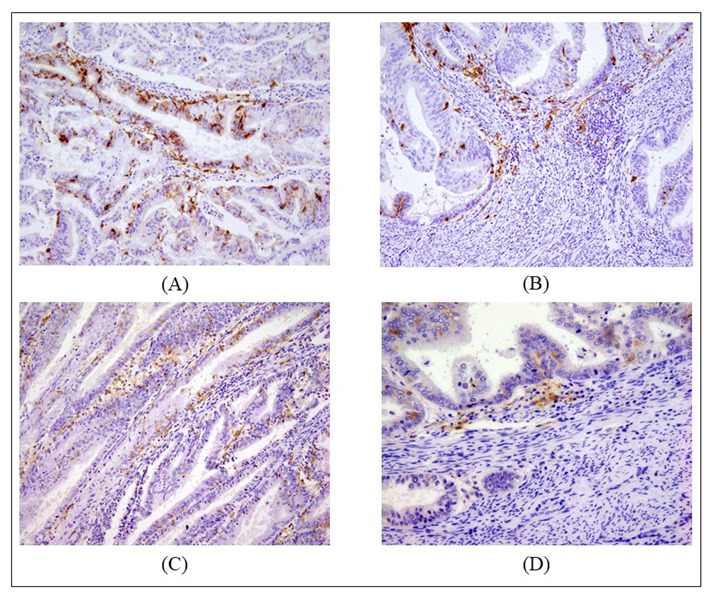
Examples of positively identified DCs infiltrating EC samples. A cell was considered as positively stained if at 40× magnification it had a visible nucleus, dendritic appearance, and intensely colored cytoplasm. (**A**) DC CD1a+ infiltrating stroma and glands of endometroid cancer at 20× magnification; (**B**) DC CD1a+ infiltrating border of endometroid cancer at 20× magnification; (**C**) DC CD1c+ infiltrating stroma and glands of endometroid cancer at 20× magnification; (**D**) DC CD1c+ infiltrating invasive border of endometroid cancer at 40× magnification (9 cells/HPF).

**Figure 4 ijms-24-01933-f004:**
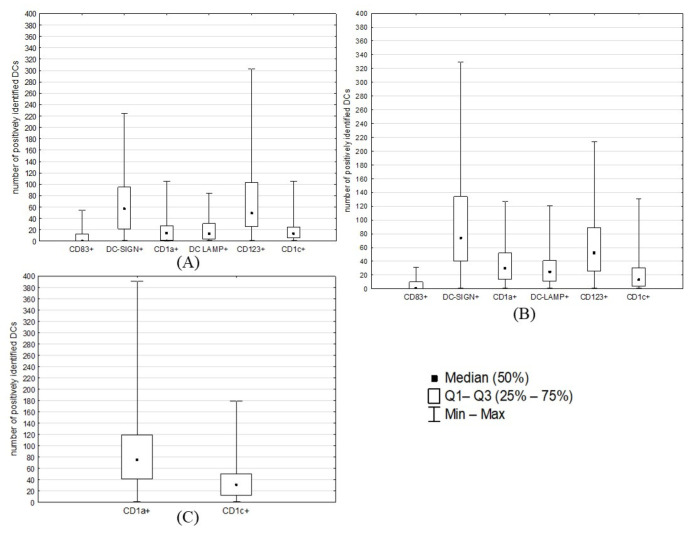
Median counts (/mm^2^) of DC subtypes in different areas of cancerous tissues: (**A**) invasive margin; (**B**) tumor stroma; (**C**) tumor glands (CD1a+ and CD1c+ DCs were the only DC subtypes found in glandular epithelium). The graph shows the median, minimal, and maximal numbers of DCs positively stained for different antibodies (counted in five fields of view) as well as values of quartile 1 (Q1) and quartile 3 (Q3) that were used to divide samples into groups for further analysis. EC samples were initially examined at 10× magnification; within areas that were most abundant in positively identified DCs, their numbers in five representative fields of view were summed up at high (40×) magnification and expressed per 1 mm^2^.

**Figure 5 ijms-24-01933-f005:**
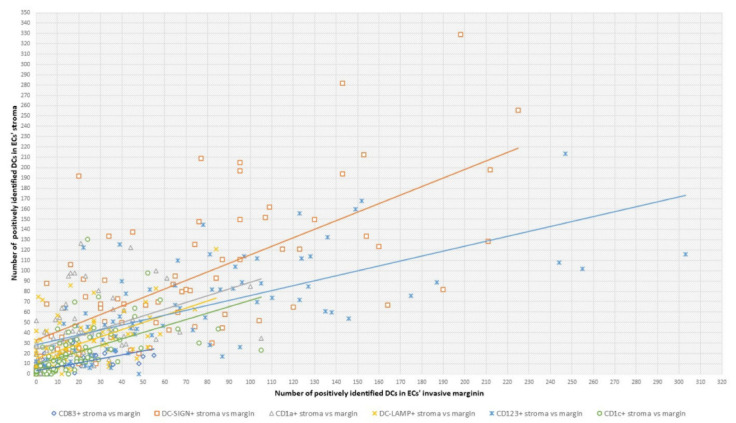
Correlations between DC counts in ECs’ stroma and invasive margin. This graph shows the numbers of DCs that were positively identified for stated antibodies in the tumor stroma (Y-axis) and invasive margin (X-axis) plotted against each other. Trend lines assigned to a pair of variables are the same color as the variables’ data points. The Spearman rank correlation test was used for statistical analysis. The obtained Spearman’s rank correlation coefficient ρ and *p*-value for every pair of correlated variables are listed below. A *p*-value < 0.05 was considered statistically significant. ◊ DC CD83+ stroma vs. margin: ρ = 0.8365, *p* < 0.0000001; □ DC SIGN+ stroma vs. margin: ρ = 0.6829, *p* < 0.0000001; ∆ DC CD1a+ stroma vs. margin: ρ = 0.5957, *p* < 0.0000001; × DC LAMP+ stroma vs. margin: ρ = 0.5278, *p* < 0.0000001; * DC 123+ stroma vs. margin: ρ = 0.7074, *p* < 0.0000001; ° DC CD1c+ stroma vs. margin: ρ = 0.7624, *p* < 0.0000001.

**Figure 6 ijms-24-01933-f006:**
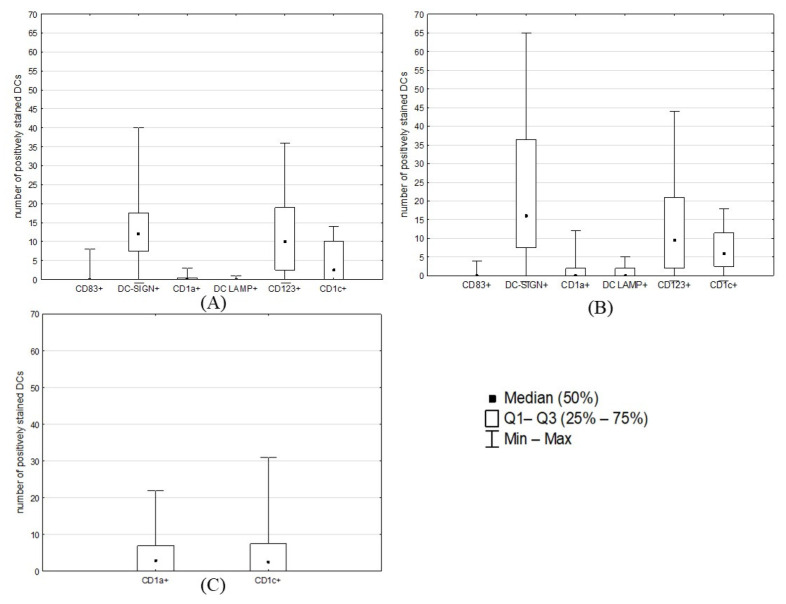
Median counts (/mm^2^) of DC subtypes in: (**A**) normal endometrial margin; (**B**) normal endometrial stroma; (**C**) in normal endometrial glands (CD1a+ and CD1c+ DCs were the only DCs subtypes found in normal glandular epithelium). The graph shows the median, minimal, and maximal numbers of DCs positively stained for different antibodies (counted in five fields of view) as well as values of quartile 1 (Q1) and quartile 3 (Q3). Areas of non-cancerous endometrium within EC slides were initially examined at 10× magnification; within areas most abundant in positively identified DCs, their numbers in five representative fields of view were summed up at high (40×) magnification and expressed per 1 mm^2^.

**Figure 7 ijms-24-01933-f007:**
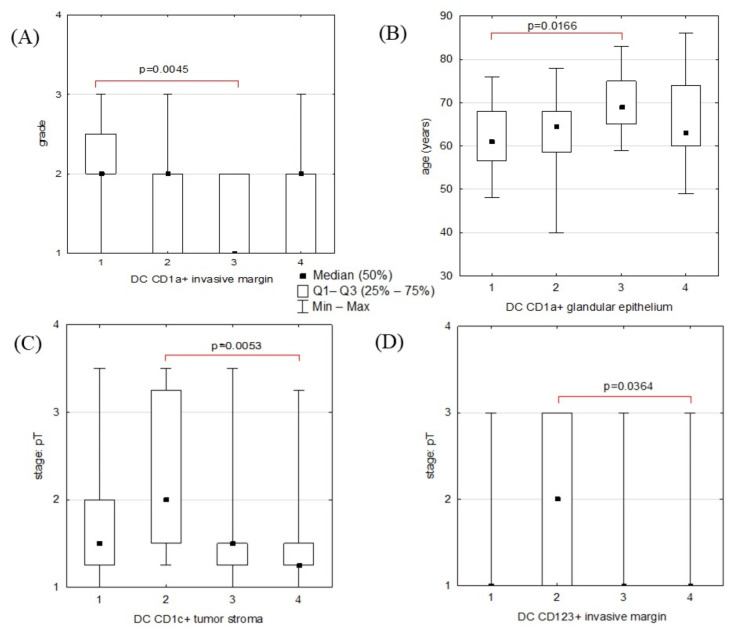
(**A**) Distribution of pathological grade scores between four groups of ECs distinguished based on the amount of DC CD1a+ present in the invasive margin; (**B**) median age (years) between four groups of ECs distinguished based on the amount of DC CD1a+ present in the glandular epithelium; (**C**) distribution of pathological stage: pT scores between four groups of ECs distinguished based on the amount of DC CD1c+ present in the stroma; (**D**) distribution of pathological stage: pT scores between four groups of ECs distinguished based on the amount of DC CD123+ present in the invasive margin. This graph shows all comparisons between the number of different DCs and clinicopathological features of ECs that were statistically significant (Table 4). The statistical analysis was performed using the Kruskal–Wallis ANOVA test (followed by the post hoc Dunn test). ECs groups significantly different in the selected dependent variable are linked by brackets captioned with the *p*-value calculated in the post hoc Dunn test. *p* < 0.05 was considered statistically significant.

**Table 1 ijms-24-01933-t001:** Patient characteristics. For this study, 94 tissue samples of malignant endometrial neoplasms were collected between 2012 and 2014 from the archives of the Department of Pathomorphology, Jagiellonian University Medical College, Krakow. Pathomorphological examination led to the identification of six different histological types. Clinicopathological features were analyzed only for endometrioid ECs (87 samples) because the number of non-endometrioid cancers was insufficient. Endometrioid carcinomas were graded with a 3-tier system developed by the International Federation of Gynecology and Obstetrics (FIGO) that counts glandular architecture, percentage of non-squamous solid components, and nuclear atypia. Pathologic TNM staging of ECs was performed according to *AJCC Cancer Staging Manual,* 7th edition, which was valid at the time of diagnosis. The depth of myometrial invasion and cervical stromal, serosal, and adnexal involvement were relevant to the pT category. Data regarding patients’ ages were collected, based on which five age groups were created.

Patients’ Characteristics			
Parameters			All patients (94)
Age, median in years (range)			64 (40–86)
Histological type, % in column (n)		Endometroid cancer	92.55 (87)
		Clear cell carcinoma	1.06 (1)
		Mucinous carcinoma	1.06 (1)
		Mixed carcinoma	1.06 (1)
		Carcinosarcoma	2.13 (2)
		Not specified	2.13 (2)
**Data for endometroid cancers only:**			
Parameters			All patients (87)
Age, median in years (range)			64 (40–86)
Age groups (y.o.), % in column (n)		1: <40; 50)	4.60 (4)
		2: <50; 60)	20.69 (18)
		3: <60; 70)	44.83 (39)
		4: <70; 80)	24.14 (21)
		5: ≥80	5.75 (5)
Stage	pT, % in column (n)	1	21.75 (4)
		1a	35.63 (31)
		1b	28.74 (25)
		2	12.64 (11)
		3a	11.49 (10)
		3b	6.90 (6)
		not reported	0 (0)
	pN, % in column (n)	0	81.61 (71)
		1	3.45 (3)
		2	1.15 (1)
		not reported	13.80 (12)
Grade, % in column (n)		1	39.08 (34)
		2	50.57 (44)
		3	9.20 (8)
		not reported	1.15 (1)

**Table 2 ijms-24-01933-t002:** Differences in numbers of different DC subtypes between ECs’ stroma and invasive margin. For DCs positively stained for DC-SIGN, CD1a, and DC-LAMP, the tumor stroma was significantly more heavily infiltrated by DCs than its invasive margin. The statistics were calculated using the Mann–Whitney U test (* *p* < 0.05 was considered statistically significant).

Compared Areas	DCs Subtype	*p*-Value *
tumor stroma vs. tumor invasive margin	DC CD83+	0.6818
	DC DC-SIGN+	**0.0278**
	DC CD1a+	**<0.00001**
	DC DC-LAMP+	**0.0061**
	DC CD123+	0.6312
	DC CD1c+	0.9522

**Table 3 ijms-24-01933-t003:** Differences in numbers of different DC subtypes between ECs and normal endometrium. For all evaluated areas and all used antibodies, EC tissues were significantly more heavily infiltrated by DCs than normal endometrium. The statistics were calculated using the Mann–Whitney U test (* *p* < 0.05 was considered statistically significant).

Compared Areas	DCs Subtype	*p*-Value *
tumor stroma vs. normal endometrial stroma	DC CD83+	**0.0016**
	DC DC-SIGN+	**<0.00001**
	DC CD1a+	**<0.00001**
	DC DC-LAMP+	**<0.00001**
	DC CD123+	**<0.00001**
	DC CD1c+	**0.0099**
tumor invasive margin vs. normal endometrial margin	DC CD83+	**0. 0155**
	DC DC-SIGN+	**<0.00001**
	DC CD1a+	**<0.00001**
	DC DC-LAMP+	**<0.00001**
	DC CD123+	**<0.00001**
	DC CD1c+	**0.0001**
tumor glands vs. normal endometrial glands	DC CD1a+	**<0.00001**
	DC CD1c+	**<0.00001**

**Table 4 ijms-24-01933-t004:** Associations between counts of different DC subtypes and clinicopathological features of endometrioid cancers (ECs). For every IHC staining, samples of ECs were assigned to four groups based on the Q1, Me, and Q3 value of the positively identified DC numbers. The table shows differences in the cancer’s grade and stage (pT and pN) between mentioned groups. The statistics were calculated using the Kruskal–Wallis ANOVA followed by Dunn’s post hoc test unless stated otherwise. Q1 and Q3—the first and the third quartile, respectively; Me—median; pT and pN—elements of pathological TNM classification established for uterine cancers; †—for DC CD83+, the statistics were calculated using the Fisher test. * *p* < 0.05 was considered statistically significant. ** Statistically significant *p*-values from Dunn’s post hoc test are placed in brackets: for DC CD1a+ margin—significant differences between groups 1 and 3, for DC CD1c+ stroma—significant differences between groups 2 and 4, for DC CD123+ margin—significant differences between groups 2 and 4.

Dependent Variable	Independent (Grouping) Variable	*p*-Value *
Grade	DC CD83+ stroma	0.7527 †
	DC DC-SIGN+ stroma	0.5165
	DC CD1a+ stroma	0.2713
	DC DC-LAMP+ stroma	0.1511
	DC CD123+ stroma	0.7394
	DC CD1c+ stroma	0.8745
	DC CD83+ margin	0.4618 †
	DC DC-SIGN+ margin	0.2071
	DC CD1a+ margin	**0.0379 (0.0045)** **
	DC DC-LAMP+ margin	0.0942
	DC CD123+ margin	0.2027
	DC CD1c+ margin	0.4012
	DC CD1a+ glands	0.5371
	DC CD1c+ glands	0.2765
Stage: pT	DC CD83+ stroma	0.5902 †
	DC DC-SIGN+ stroma	0.1281
	DC CD1a+ stroma	0.2415
	DC DC-LAMP+ stroma	0.9356
	DC CD123+ stroma	0.7739
	DC CD1c+ stroma	**0.0054 (0.0053)** **
	DC CD83+ margin	0.1669 †
	DC DC-SIGN+ margin	0.0670
	DC CD1a+ margin	0.8728
	DC DC-LAMP+ margin	0.0705
	DC CD123+ margin	**0.0037 (0.0364)** **
	DC CD1c+ margin	0.4080
	DC CD1a+ glands	0.4124
	DC CD1c+ glands	0.6116
Stage: pN	DC CD83+ stroma	0.3070 †
	DC DC-SIGN+ stroma	0.9977
	DC CD1a+ stroma	0.1801
	DC DC-LAMP+ stroma	0.6962
	DC CD123+ stroma	0.5843
	DC CD1c+ stroma	0.4771
	DC CD83+ margin	0.7191 †
	DC DC-SIGN+ margin	0.5881
	DC CD1a+ margin	0.9788
	DC DC-LAMP+ margin	0.6839
	DC CD123+ margin	0.6076
	DC CD1c+ margin	0.5913
	DC CD1a+ glands	0.0725
	DC CD1c+ glands	0.4004

**Table 5 ijms-24-01933-t005:** Criteria for allocation of EC samples to groups based on the numerosity of different DC subtypes infiltrating cancerous tissues. These criteria were uniform for each antibody used for IHC staining: the first group included samples with DC counts less than the calculated number of the first quartile (Q1); the second group included DC counts greater than or equal to Q1 and less than the median value (Me); the third group included DC counts greater than or equal to Me and less than the third quartile (Q3); the fourth group included DC counts greater than Q3. ***** There was an exception for staining with the anti-CD83 antibody. The Q1 was 0 for the number of DCs 83+ in stroma and border, so four groups could not be distinguished. Two groups were created instead based on the Me value (the first group contained samples with DC counts less than or equal to Me; the second group included DC counts greater than Me). Rows with missing data include samples in which the numbers of DCs could not be established due to tissue damage during slide preparation or inappropriate IHC staining.

DCs Subtype	Criteria for Group Allocation	No of Samples
DC CD83+ stroma *	1st group: No DCs ≤ 1	41
	2nd group: No DCs >1	43
	missing data	3
DC CD83+ margin *	1st group: No DCs ≤ 0	48
	2nd group: No DCs > 0	36
	missing data	3
DC DC-SIGN+ stroma	1st group: No DCs < 40	20
	2nd group: 40 ≤No DCs < 74	20
	3rd group: 74 ≤No DCs < 134	19
	4th group: No DCs ≥ 134	21
	missing data	7
DC DC-SIGN+ margin	1st group: No DCs < 22	19
	2nd group: 22 ≤No DCs < 57	19
	3rd group: 57 ≤No DCs < 95	17
	4th group: No DCs ≥ 95	22
	missing data	10
DC CD1a+ stroma	1st group: No DCs <14	19
	2nd group: 14 ≤No DCs < 30	21
	3rd group: 30 ≤No DCs < 52	20
	4th group: No DCs ≥ 52	21
	missing data	6
DC CD1a+ margin	1st group: No DCs < 2	16
	2nd group: 2 ≤No DCs < 14.50	24
	3rd group: 14.50 ≤No DCs < 27	19
	4th group: No DCs ≥ 27	21
	missing data	7
DC CD1a+ glands	1st group: No DCs < 41	20
	2nd group: 41 ≤No DCs < 75	20
	3rd group: 75 ≤No DCs < 119	18
	4th group: No DCs ≥ 119	23
	missing data	6
DC LAMP+ stroma	1st group: No DCs < 11	20
	2nd group: 11 ≤No DCs < 25	20
	3rd group: 25 ≤No DCs < 41	22
	4th group: No DCs ≥ 41	21
	missing data	4
DC LAMP+ margin	1st group: No DCs < 4	18
	2nd group: 4 ≤No DCs < 14	20
	3rd group: 14 ≤No DCs < 31	23
	4th group: No DCs ≥ 31	22
	missing data	4
CD123+ stroma	1st group: No DCs < 25.50	21
	2nd group: 25.50 ≤No DCs < 52.50	21
	3rd group: 52.50 ≤No DCs < 88.50	21
	4th group: No DCs ≥ 88.50	21
	missing data	3
CD123+ margin	1st group: No DCs < 26	21
	2nd group: 26 ≤No DCs < 49	21
	3rd group: 49 ≤No DCs < 103	21
	4th group: No DCs ≥ 103	22
	missing data	3
CD1c+ stroma	1st group: No DCs < 4	20
	2nd group: 4 ≤No DCs < 13	21
	3rd group: 13 ≤No DCs < 30	20
	4th group: No DCs ≥ 30	22
	missing data	4
CD1c+ margin	1st group: No DCs < 6	19
	2nd group: 6 ≤No DCs < 14	22
	3rd group: 14 ≤No DCs < 25	21
	4th group: No DCs ≥ 25	21
	missing data	4
CD1c+ glands	1st group: No DCs < 12	20
	2nd group: 12 ≤No DCs < 31	18
	3rd group: 31 ≤No DCs < 50.50	22
	4th group: No DCs ≥ 50.50	20
	missing data	7

## Data Availability

Data supporting the reported results can be found under links in the Appendix A. Other data presented in this study are available upon request from the corresponding author.

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
