# Peer review of "Distribution of DC Subtypes: CD83+, DC-LAMP+, CD1a+, CD1c+, CD123+, and DC-SIGN+ in the Tumor Microenvironment of Endometrial Cancers—Correlation with Clinicopathologic Features"

_ijms, 2023, doi:10.3390/ijms24031933_

Round 1

Reviewer 1 Report

In the manuscript titled “Distribution of DCs subtypes: CD83+, DC-LAMP+, CD1a+, CD1c+, CD123+, and DC-SIGN+ in the tumor microenvironment of endometrial cancers - correlation with clinicopathologic features”, the authors immunohistochemically analyzed 94 patient samples of malignant endometrial neoplasms, aiming to assess different subsets of dendritic cells in endometrial cancers samples. The research is a nice attempt to find the correlation between DCs' infiltrates and clinicopathological features of endometrial cancers, yet the current version needs further improvement for publication. 

1.     Please specify the definition of DCs scores.

2.   Line 133, unrelated content: “this section may be divided by subheadings. It should provide a concise and precise description of the experimental results, their interpretation, as well as the experimental conclusions that can be drawn.”

3.    “In further analyses, we compared groups created for each IHC staining based on median, Q1, and Q3 values of DCs count.” The authors should define the criteria precisely.

4.   “We found statistically significant differences in grade value distribution between four groups of DCs CD1a+ present in cancer's invasive margin [Figure 6].” What information do these differences provide to us? Why are they different with each other? The authors should provide detailed analysis on these results.

5.    “We observed some diversity in patients' age between four groups of DC CD1a+ present in tumors' glandular epithelium, with the youngest patients represented predominantly in the group of the least numerous DCs [Figure 7].” It appears that the highest  DC CD1a+ glandular epithelium group has relatively young age. How to explain?

6.    Interestingly, cancers with the least DC CD1c+ present in the stroma (DCs group 2) infiltrated surrounding tissues to a greater degree than ones most heavily infiltrated by CD1c positive cells (DCs group 4) [Figure 8]. Similarly, cancers with the highest observed pT were predominant in the DC CD123+ group 2 [Figure 9].” What do these results indicate? The authors should provide deeper analysis on these observations.

Author Response

  1. Please specify the definition of DCs scores.

The term "DCs scores" was improperly used in a sentence "Some papers prove that high DCs scores are related to better survival, while others claim that the increasing DCs infiltration is positively associated with cancer stage, grade, and poor outcomes." (Line 49). It will be replaced with the more accurate term "DCs counts" defined as the number of positively stained DCs in assessed areas of cancerous tissues. 

  1. Line 133, unrelated content: “this section may be divided by subheadings. It should provide a concise and precise description of the experimental results, their interpretation, as well as the experimental conclusions that can be drawn.”

This line was supposed to be removed from the Microsoft Word template used for the manuscript preparation. Unfortunately, it was not removed by the oversight. It will not be included in the final version of the article.

  1. “In further analyses, we compared groups created for each IHC staining based on median, Q1, and Q3 values of DCs count.” The authors should define the criteria precisely.

The criteria determined for creating groups were uniform for each antibody used for IHC staining. The first group included samples with DCs counts less than the calculated number of the first quartile (Q1); the second group: with DCs counts greater than or equal to Q1 and less than the median value (Me); the third group: with DCs counts greater than or equal Me and less than the third quartile (Q3); the fourth group: with DCs counts greater than Q3. There was an exception for staining with antibody anti-CD83. The Q1 was 0 for the number of DCs 83+ in stroma and border, so allocation to four groups was unfounded. Two groups were created instead, based on the Me value (the first group: samples with DCs counts less than or equal to Me; the second group: with DCs counts greater than Me). We added descriptions of precise criteria used for dividing ECs samples into groups based on the numerosity of different DCs subtypes [Table 5.] in the corrected version of the article.

  1. “We found statistically significant differences in grade value distribution between four groups of DCs CD1a+ present in cancer's invasive margin [Figure 6].” What information do these differences provide to us? Why are they different with each other? The authors should provide detailed analysis on these results.

Indeed, the Kruskal-Wallis ANOVA test followed by Dunn’s post hoc test showed a significant difference between group 3 and 1 of DCs CD1a+ present in the invasive margin. Cancers with the least numerous DCs CD1a+ on their infiltrative borders had statistically higher grade scores than cancers with higher DCs counts. The above results show associations between lower numbers of immature DCs CD1a+ and greater histological malignancy. We have added a more detailed analysis of these results to the Discussion section (Line 326 - 332): “In this study, cancers with the least numerous DCs CD1a+ on their infiltrative borders had statistically higher grade scores than ones with higher DCs counts. This observation can result from the fact that ECs with higher grade scores are less likely to form glandular structures, to which DCs CD1a+ showed great affinity. It is also possible that a higher number of DCs CD1a+ in cancerous glands induces effective immune responses that control ECs progression and dedifferentiation, hence lower grade scores.

  1. “We observed some diversity in patients' age between four groups of DC CD1a+ present in tumors' glandular epithelium, with the youngest patients represented predominantly in the group of the least numerous DCs [Figure 7].” It appears that the highest DC CD1a+ glandular epithelium group has relatively young age. How to explain?

In our study, the youngest patients were represented predominantly in the ECs group of the least numerous DC CD1a+ present in tumors' glandular epithelium. Patients' age in this group was significantly lower than in cancers with higher counts of CD1a-positive cells in their glands. Literature shows that aging is related to alterations in the structure and function of the immune system, including cell-intrinsic changes in innate immune cells (Goronzy and Weyand, 2013; Amir A. Sadighi Akha, 2018). Older age is associated with higher concentrations of pro-inflammatory cytokines that affect the activation and maturation of DCs. However, it is also directly related to a significantly lower density of DC CD1a+ in mucosal or skin epithelium (Shurin et. al., 2007). "Aged DCs" display partially activated phenotype (higher basal levels of NF-κB activation without simultaneous upregulation of CD86 or CD80) and are more reactive to self-antigens. A lower expression of CD1a+ was expected in older patients. Yet in the ECs groups 1 to 3, there was a tendency for DC CD1a+ numerosity to increase along with age. Contrary, cancers with the most numerous DC CD1a+ (group 4) in the glandular epithelium comprised relatively young patients. An inconsistency between these two observations may result from small group sizes or large diversity of patients' age in four groups. The age cut-off point for the elderly is contractual, so the median age in all four groups could be classified as old. The above could explain inconclusive correlations between DCs numerosity and age. Studies comparing elderly and non-elderly patients managed to obtain more unequivocal results. Another explanation is the possible difference in percentages of glandular epithelium - linked with DC CD1a+ numerosity - between cancers from different groups. Yet, there were no statistically significant differences in ECs grade (or stage) between patients of different ages, so clinicopathological features can not fully explain the observed variation of CD1a+ expression between the groups.

  1. “Interestingly, cancers with the least DC CD1c+ present in the stroma (DCs group 2) infiltrated surrounding tissues to a greater degree than ones most heavily infiltrated by CD1c positive cells (DCs group 4) [Figure 8]. Similarly, cancers with the highest observed pT were predominant in the DC CD123+ group 2 [Figure 9].” What do these results indicate? The authors should provide deeper analysis on these observations.

Regarding: “Interestingly, cancers with the least DC CD1c+ present in the stroma (DCs group 2) infiltrated surrounding tissues to a greater degree than ones most heavily infiltrated by CD1c positive cells (DCs group 4) [Figure 8].”These results show that cancers with higher counts of intrastromal DCs CD1c+ tend to have a lower pathological stage and are less likely to infiltrate surrounding tissues than ECs with a lower number of DC CD1c+. It could indicate that cDC2s (characterized by CD1c+ expression), despite being less potent APCs than cDC1s, still associate with favorable features of ECs and could be related to a better prognosis. However, more data is needed to verify these statements. As mentioned in the Discussion section, the capacity of cDC2s to migrate to LNs and drive effective CD4+ T-cell responses may be the reason for their favorable impact.

Regarding: “Similarly, cancers with the highest observed pT were predominant in the DC CD123+ group 2 [Figure 9].” We observed that the ECs group with the highest infiltration of DC CD123+ in invasive borders (group 4) had significantly more cases with lower cancerous infiltration depth compared to a group with decreased DC CD123+ density (group). These results suggest that plasmacytoid DCs (characterized by CD123 expression), despite lower antigen cross-presentation capacity, could facilitate effective antitumor responses that control infiltration and metastases, features assessed while creating pT scores. In the literature, plasmacytoid DCs were described to contribute to antitumor responses to a lesser degree due to impaired responses to TLR activation. Contrary results obtained in this study could be an argument for the recently brought up thesis that potentially not all cancers cause pDCs hypofunction. And that this DCs subtype can control cancer growth by IFNα secretion, which is essential for cDC1 maturation and stimulation of local CD8+ T-cells.

Please find all corrections introduced to the article in the attachment.

Reviewer 2 Report

6: Authors: It lists 3 of the 4 authors as the chair of pathomorphology but I believe only Joanna is chair. Also I can’t find Apolonia as a faculty member at Jagiellonian U. but this could of course be my own oversight

16: “positive DC’s” is confusing, maybe ‘positively identified’

51: It feels like these few sentences with your justification and hypothesis (which I really like by the way) should be at the end. I realize you introduce the markers here as part of that hypothesis, but the way the intro is ending just isn’t flowing well into the results

Results:

-I am confused about your counting method. I understand that you are staining tissues against multiple markers, ostensibly because there is confusion in the literature regarding what markers correlate to either positive or negative patient outcomes. But tissues aren’t being stained simultaneously for multiple markers, so how can you prove that any of these cells are actually dendritic cells? I mean, I get that some of these markers are pretty solid DC markers, but CD1c could be monocytes. You even mention this in the discussion (line 215), and talk about how there could be off-target staining of non-immune cells.

-There is also not enough description in the results. Figures 4 and 5 only have one sentence, as do many other figures in the text, and the figure descriptions themselves are inadequate in their details and statistics. Each figure should be able to stand alone. A lot of these figures should actually be combined and labelled with A, B, C, etc., especially those that correlate together, that would help tell your overall story.

-Are these cancer severity gradations all equal for the first 6 figures?

-You should have example histology images, including an exact description of the markers stained in each image, alongside an aggregate count of all samples in graph form. You have lots of examples of both, but it is very difficult to confirm your conclusions when images are buried in the supplementary files (and those supplements are not very well organized).

-A lot of your supplementary data should be in the paper itself. If we’re trying to show which markers actually correlate to which outcomes, describing which markers DON’T correlate should be equally important, if not more so, because other researchers who might rely on those markers might question your findings.

- 133-136: It seems like this is from a previous reviewer?

-Figures 10-12: these are interesting ways to display these data. It seems like it would be easier to show that these markers are correlating with specific cancer grades by just running an ANOVA instead of a Spearman. It just seems off to have the slope partially tied to the 5 possible grades.

-The discussion has a lot of great review information, but is very long. A lot could be dropped to bring more focus back to your story. I think you could actually turn most of it into a review article on DC subsets in EC.

Materials and Methods: These need to be sectioned off by experiment.

-And for the staining, I get that your department has standards, but maybe mention that you follow typical protocols in this field.

-Who actually performed the pathology analysis?

-The methods should be described less personally (less “we”), and should be more methodical, if you will

-It is confusing that there are figures in the appendix, AND supplemental figures

Author Response

6: Authors: It lists 3 of the 4 authors as the chair of pathomorphology but I believe only Joanna is chair. Also I can’t find Apolonia as a faculty member at Jagiellonian U. but this could of course be my own oversight

The term chair of pathomorphology was used as a synonym for the Department of Pathomorphology. This way of describing affiliation was used in previous publications by department employees. Indeed the phrase chair could be unclear in this context. Dr. Joanna Szpor and Dr. Grrzegorz Dyduch work in the Department of Pathomorphology at Jagiellonian University. Apolonia Miążek is a fifth-year medical student at Jagiellonian University Medical College and a member of the Students Scientific Group under the Department of Pathomorphology, hence the decision to use the same affiliation for Apolonia as for other members of this department. If needed, we can deliver a certificate confirming that Apolonia is a member of this Students Scientific Group.

16: “positive DC’s” is confusing, maybe ‘positively identified’

That is a valid point. The term positive DCs was changed to positively identified DCs and positively stained DCs to avoid confusion.

51: It feels like these few sentences with your justification and hypothesis (which I really like by the way) should be at the end. I realize you introduce the markers here as part of that hypothesis, but the way the intro is ending just isn’t flowing well into the results

We agree with your opinion that the transition from the end of the Introduction to the Results is not very smooth. Sentences from lines 57 - 70 were transferred to the end of the article (Discussion section).

-I am confused about your counting method. I understand that you are staining tissues against multiple markers, ostensibly because there is confusion in the literature regarding what markers correlate to either positive or negative patient outcomes. But tissues aren’t being stained simultaneously for multiple markers, so how can you prove that any of these cells are actually dendritic cells? I mean, I get that some of these markers are pretty solid DC markers, but CD1c could be monocytes. You even mention this in the discussion (line 215), and talk about how there could be off-target staining of non-immune cells.

Indeed, in the design of this study, simultaneous staining for multiple markers was not planned. Assessment of that many stainings present in one sample could presumably be difficult, as all identified molecules are cytoplasmic and could be easily mistaken for each other, and the IHC results would be unclear. In the Department of Pathology, we usually only perform simultaneous stainings for up to two different antibodies. Certainly, detecting more than one DCs marker on an assessed cell would make identification more reliable and specific. Despite using individual stainings, we made all effort to ensure that cells included in the calculation are DCs. Often areas of the highest DCs density within a tissue slice were the same for most of the used antibodies. We could verify whether DCs positively identified for one antibody were also positive for other markers in the same hot spot. Cells' morphology was assessed before establishing their numerosity. We only counted cells with a visible nucleus and dendritic appearance, e.g. the presence of cytoplasmic processes (except for CD123+ plasmacytoid DCs). We are aware that there is still a risk that non-immune cells were falsely classified as DCs. However, separate samples assessment by two authors aimed to minimalize potential bias.

-There is also not enough description in the results. Figures 4 and 5 only have one sentence, as do many other figures in the text, and the figure descriptions themselves are inadequate in their details and statistics. Each figure should be able to stand alone. A lot of these figures should actually be combined and labelled with A, B, C, etc., especially those that correlate together, that would help tell your overall story.

We agree with your suggestions. The corrected article will include more detailed and organized descriptions of our results. All findings, including statistically insignificant ones, will be presented [Table 2., Table 2., Table S2.]. Similarly, figures will be combined whenever possible and justified, and captions corrected. Please find all introduced corrections in the attached version of the article.

-Are these cancer severity gradations all equal for the first 6 figures?

We are not sure if we understand your question. Examples of histology images include cancers evaluated as grade 1 or grade 3: Figure 1 – G1 (left) and G3 (right), Figure 4 – G1 for cancers on both images. Figures 2., 3., and 5. show median counts of DCs subsets in cancer stroma, margin, and glands, respectively. Data collected for all endometrioid cancers were presented in these three graphs, and cancers differed in their grade and stage values [Table 1]. Figure 6. shows the distribution of pathological grade scores between four cancer groups divided based on the number of DCs CD1a+ present in the invasive margin. Figures added to the newest version of the article [Figure 1. – 3.] represent cancers of all possible grade values (1, 2, and 3). Does our answer exhaust your question?

-You should have example histology images, including an exact description of the markers stained in each image, alongside an aggregate count of all samples in graph form. You have lots of examples of both, but it is very difficult to confirm your conclusions when images are buried in the supplementary files (and those supplements are not very well organized).

As you suggested, we have added representative histology images for all antibodies to the main text  [Figure 1. – 3.]. Examples of stained markers were described in more detail in the captions below the figures. You can find propositions of graphs illustrating aggregate counts of DCs stained for all antibodies, separately for tumor stroma, border, and glands, in the Supplementary Materials [Figure S2. and S3].

-A lot of your supplementary data should be in the paper itself. If we’re trying to show which markers actually correlate to which outcomes, describing which markers DON’T correlate should be equally important, if not more so, because other researchers who might rely on those markers might question your findings.

According to your suggestion, all results were added to the main text, including these not statistically significant [Table 2. – 4.]. Similarly, valid data from the Supplementary Material section were transferred to the main text [Figure 1. – 3., Figure 5., Table 2.-5.].

- 133-136: It seems like this is from a previous reviewer?

We forgot to remove this line from the Microsoft Word template used for the manuscript preparation by the oversight. We will not include it in the final version of the article.

Results:

-Figures 10-12: these are interesting ways to display these data. It seems like it would be easier to show that these markers are correlating with specific cancer grades by just running an ANOVA instead of a Spearman. It just seems off to have the slope partially tied to the 5 possible grades.

We agree with your suggestion. ANOVA Kruskall-Wallis test was performed for these data and presented in Table. 4. We have removed Graphs showing Spearman correlations from the main text as they did not bring any new information compared to ANOVA.

-The discussion has a lot of great review information, but is very long. A lot could be dropped to bring more focus back to your story. I think you could actually turn most of it into a review article on DC subsets in EC.

We will try to shorten the discussion without losing our main points.

Materials and Methods: These need to be sectioned off by experiment.

-And for the staining, I get that your department has standards, but maybe mention that you follow typical protocols in this field.

-Who actually performed the pathology analysis?

-The methods should be described less personally (less “we”), and should be more methodical, if you will

We agree that describing our methodology section by section would be more transparent to the reader and applied it to the article. We added a more detailed description of the methodology. We also included a notification that all used protocols stay in line with typical protocols in the field. Please find all introduced corrections in the attached version of the article.

Two authors, Dr. Grzegorz Dyduch and Apolonia Miążek performed pathological analysis. Cells were counted separately by two authors, and if disagreements occurred, they were solved by discussion.

-It is confusing that there are figures in the appendix, AND supplemental figures

We will delete the Appendix section and transfer all side data to the Supplementary Materials section.

Please find all corrections introduced to the article in the attachment.

Round 2

Reviewer 1 Report

The reviewer agrees on the authors' responses.